# Screening of In-Vitro Anti-Inflammatory and Antioxidant Activity of *Sargassum ilicifolium* Crude Lipid Extracts from Different Coastal Areas in Indonesia

**DOI:** 10.3390/md19050252

**Published:** 2021-04-28

**Authors:** Puspo Edi Giriwono, Diah Iskandriati, Nuri Andarwulan

**Affiliations:** 1Department of Food Science and Technology, Faculty of Agricultural Engineering and Technology, IPB University (Bogor Agricultural University), West Java 16680, Indonesia; saraswati_231@apps.ipb.ac.id (S.); pegiriwono@apps.ipb.ac.id (P.E.G.); 2Southeast Asian Food and Agricultural Science Technology (SEAFAST) Center, IPB University (Bogor Agricultural University), West Java 16680, Indonesia; 3Primate Research Center, IPB University (Bogor Agricultural University), West Java 16151, Indonesia; atie@indo.net.id

**Keywords:** anti-inflammatory, antioxidant, coastal area, crude lipid extract, *Sargassum ilicifolium*

## Abstract

*Sargassum* brown seaweed is reported to exhibit several biological activities which promote human health, such as anticancer, antimicrobial, antidiabetic, anti-inflammatory, and antioxidant activity. This study aimed to investigate the anti-inflammatory and antioxidant activity of crude lipid extracts of *Sargassum ilicifolium* obtained from four different coastal areas in Indonesia, namely Awur Bay–Jepara (AB), Pari Island–Seribu Islands (PI), Sayang Heulang Beach–Garut (SHB), and Ujung Genteng Beach–Sukabumi (UGB). Results showed that treatment of RAW 264.7 macrophage cells with UGB and AB crude lipid extracts (12.5–50 µg/mL) significantly suppressed the nitric oxide production after lipopolysaccharide stimulation, both in pre-incubated and co-incubated cell culture model. The anti-inflammatory effect was most marked in the pre-incubated cell culture model. Both two crude lipid extracts showed 2,2-diphenyl-1-picrylhydrazyl radical scavenging activity and high ferric reducing antioxidant power, which were amounted to 36.93–37.87 µmol Trolox equivalent/g lipid extract and 681.58–969.81 µmol FeSO_4_/g lipid extract, respectively. From this study, we can conclude that crude lipid extract of tropical *S. ilicifolium* can be further developed as a source of anti-inflammatory and antioxidant agent.

## 1. Introduction

As one of the countries with the longest coastline in the world, Indonesia’s seaweed- producing potential is very high. Indonesia was able to produce up to 10.4 million tons of fresh seaweed in 2017. In international trade, Indonesia is one of the main players, with an export volume reaching 212,962 tons in 2018 [1]. Of the global production of cultured seaweed, Indonesia contributed almost 38% to the total production, which made Indonesia the second-largest seaweed producer in the world after China [2]. Among hundreds of seaweed species found in Indonesian waters, only a few species have been optimally cultivated, especially from agar- and carrageen-producing groups such as *Gracilaria* sp., *Kappaphycus alvarezii*, and *E. denticulatum.* The utilization of brown seaweed in Indonesia is still limited, even though its health benefits have been empirically and scientifically recognized [3].

Brown seaweed, especially *Sargassum* species, has become one of the important ingredients in the traditional medicine of East Asian communities. Some *Sargassum* species have been utilized for generations by Chinese and Korean people to treat several inflammation-associated health problems, such as the painful scrotum, edema, liver organ swelling, chronic bronchitis, etc. [4,5]. Several studies on *Sargassum* summarized by Saraswati et al. [6] indicated the promising anti-inflammatory effect of this seaweed genus, for both acute and chronic conditions. Although several screening studies demonstrated the predominance of *Sargassum* lipid-soluble fractions which are responsible for inflammatory activity [7,8,9,10], the potency of the lipid-soluble compounds of tropical *Sargassum* has not been fully explored. To date of our knowledge, the sulfated polysaccharides of tropical *Sargassum* are the most studied compounds for their anti-inflammatory activity [5]. The utilization of tropical *Sargassum* lipid-soluble fractions/compounds as a source of anti-inflammatory and antioxidant agents can be one of the valorization efforts of seaweed metabolites [11].

One of the *Sargassum* species which is widely distributed in Indonesian waters with promising biomass is *S. ilicifolium*. Together with other species from *Sargassum* subgenus, this seaweed is included as one of the most abundant species distributed in the Pacific basin, primarily southwestern Pacific regions such as North Australia, Fiji, Guam, Indonesia, Malaysia, Micronesia, New Caledonia, Papua New Guinea, Philippines, Singapore, Solomon Islands, and Taiwan [12,13]. *S. ilicifolium* was first described by Agardh in 1820, and this seaweed is known from not only the tropical Pacific region, but also the Indian ocean region [14]. In addition, *S. cristaefolium, S. duplicatum, S. berberifolium*, and *Fucus latifolius* are all recognized as synonyms of *S. ilicifolium,* according to Guiry and Guiry [12]. Wu et al. [15] have reported the anti-inflammatory effect of crude sulfated polysaccharides derived from *S. ilicifolium* (as *S. cristaefolium*) in lipopolysaccharide (LPS)-induced murine macrophage RAW 264.7 cells. The anti-inflammatory studies of *S. ilicifolium* and *S. duplicatum* have been previously investigated by several research groups on different models, such as carrageenan-induced rat paw edema [16], 12-O-techanoyl-13-myristate-induced polymorphonuclear leukocyte [17], LPS-induced RAW 264.7 cells [18,19], and indomethacin-induced rat inflammatory bowel disease [20].

In the present study, we aimed to investigate the anti-inflammatory and antioxidant activity of crude lipid extracts of *S. ilicifolium* obtained from four different coastal areas around Java Island–Indonesia, namely Sayang Heulang Beach/SHB, Ujung Genteng Beach/UGB, Pari Island/PI, and Awur Bay/AB-Jepara. SHB and UGB are located at the southern waters of Java Island, while PI and AB are in the northern waters of Java. The anti-inflammatory effect of brown seaweed crude lipid extracts was assayed on LPS-induced RAW 264.7 cells under two different experimental models, i.e., pre-incubated and co-incubated model. While the antioxidant activity was examined through 2,2-diphenyl-1-picrylhydrazyl (DPPH) radical scavenging method and ferric reducing antioxidant power (FRAP) method. These study results are expected to provide baseline information for further development of Indonesian *S. ilicifolium* as a source of anti-inflammatory and antioxidant ingredients.

## 2. Results

### 2.1. Cell Viability after Treatment of S. ilicifolium Crude Lipid Extract

Cell viability assay was carried out to determine doses of samples for anti-inflammatory activity assay. Figure 1 shows RAW 264.7 cells viability after brown seaweed lipid extract treatment for 24 h. Because decreased viability was most marked at 100 µg/mL, doses of samples ranging from 12.5–50 µg/mL were chosen for further anti-inflammatory assay. Treatment of RAW 264.7 cells with crude lipid extract from Pari Island (PI) samples exhibited the highest cytotoxicity. PI extract yielded 63.9% viability at 12.5 µg/mL, while 100 µg/mL of PI extract caused a significant cytotoxic effect with viability value at 22.6%. The weakest cytotoxic effect was found in Awur Bay (AB) sample treatment with cell viability at 12.5, 25, and 50 µg/mL were 90.0, 88.8, and 74.6%, respectively. When excluding PI treatment due to its highest cytotoxic effect, UGB treatment showed a sharp decrease in viability at a dose range of 12.5–50 µg/mL. At 50 µg/mL, viability of the UGB-treated group only reached 48.22%, while the viability of SHB and AB treatment group at a dose of 50 µg/mL were 69.01 and 74.64%, respectively. Determination of cell viability becomes important, so the observed anti-inflammatory effect is deemed not to be attributable to cytotoxic effects [21,22].

### 2.2. Anti-Inflammatory Activity of S. ilicifolium Crude Lipid Extract

In this study, lipopolysaccharide (LPS) from *Escherichia coli* O111:B4 was used to stimulate an inflammatory response in RAW 264.7 macrophage cells. It was proven by the significant increase of nitric oxide (NO) production in the LPS-treated group (control +) if compared to the non-LPS treatment group (control-). Nitric oxide (NO) is a signaling molecule that plays a key role in the pathogenesis of inflammation. The impaired production of NO is associated with tissue damage, neurodegeneration, and inflammatory disorders in joint, gut, and lung since it acts as pro-inflammatory mediator which can amplify the inflammatory response. Hence, an effort to find selective NO inhibitors becomes a therapeutic advance in the management of inflammatory diseases [23].

Effects of brown seaweed lipid extract treatment on NO production of LPS-induced RAW 264.7 macrophage cells are shown in Figure 2 and Figure 3. UGB sample provided the strongest NO inhibition activity. In the pre-incubated cell culture model, level of NO inhibition by UGB treatment reached 83.21% at a dose of 50 µg/mL and 26.10% at 12.5 µg/mL. While NO inhibition level by UGB treatment in the co-incubation model ranged from 28.07–61.81%. Treatment of RAW 264.7 cells by SHB samples in the pre-incubated cell culture model exhibited no significant reversion of LPS-induced NO production. In the co-incubation model, SHB treatment gave lowest NO inhibition activity, only ranging from 10.15–25.38%. AB treatment showed a promising NO inhibition activity with a more acceptable cytotoxic effect. Level of NO inhibition by AB treatment was ranging from 11.29–65.76% in pre-incubated cell culture model and 13.44–41.80% in co-incubation model. Jaswir et al. [18] found that treatment of RAW 264.7 cells with water extracts (50 µg/mL) of three different Malaysian *Sargassum* species could inhibit LPS-induced NO production around 40–52%. A study by Jayawardena et al. [9] reported that LPS-induced NO production in RAW 264.7 cells was inhibited by treatment of Korean *S. horneri* ethanolic extract through suppression of nuclear factor (NF)-kappa B transactivation, decreased mitogen-activated protein kinases (MAPKs) phosphorylation, and increased expression of nuclear factor E2-related factor 2 (Nrf2) and heme oxygenase 1 (HO-1). The activation of NF-kB transcription factor and MAPKs (i.e., extracellular-signal regulated kinase (ERK), Jun N-terminal kinase (JNK), and p38) are known to play a critical role in the regulation of inflammatory response, whilst the activation Nrf2/HO-1 pathway induce the cytoprotective effect against oxidative stress due to inflammatory stimulation.

Pre-incubation of RAW 264.7 cells with *S. ilicifolium* lipid extracts before LPS stimulation exerted a stronger anti-inflammatory effect than in co-incubation mode. This indicated that bioactive compounds contained in the tested extracts could counter the inflammatory response through intracellular actions due to their capability in permeating cell membranes. Thus, the outcome of pre-incubation may reflect a preventive action [24]. Gany et al. [25] revealed that pre-incubation of C8B4 microglia cells for 3 h with lipid-soluble extract (0.4 mg/mL) of *Padina australis* was able to significantly reduce LPS-induced NO production up to 75.67%. Moreover, LPS-induced pro-inflammatory cytokines production (e.g., tumor necrosis factor (TNF)-α, interleukin (IL)-6, and IL-1β) of those cells were also reversed by the aforementioned treatment. On the contrary of our results, Hidalgo et al. [26] found that co-incubation of RAW 264.7 macrophages cells with protocatechuic acid (phenolic compound) and inflammatory stimulant (LPS and/or IFN-γ) gave stronger NO inhibition than in the pre-incubation model. The anti-inflammatory effect of phenolic compounds might be generated from their radicals scavenging activity in the surrounding environment of inflamed cells [27] and direct interaction of phenolic compounds with inflammatory stimulants [26].

### 2.3. DPPH Radical Scavenging and Ferric Reducing Ability of S. ilicifolium Crude Lipid Extract

Since oxidative stress and inflammation are closely linked and affecting each other, exploring anti-inflammatory and antioxidant properties of natural compounds becomes an interesting research topic [28]. Anti-inflammatory and antioxidant activity of natural compounds may contribute to the prevention of chronic diseases, e.g., cardiovascular diseases, cancer, diabetes, Alzheimer’s, etc. [29]. In the present study, the antioxidant activity of brown seaweed lipid extract was assessed through the DPPH radical scavenging and FRAP assay (Table 1).

SHB sample showed the lowest DPPH scavenging activity, while PI and SHB sample exhibited the lowest ferric reducing antioxidant power per gram dry lipid extract. On the other hand, AB and SHB samples had the higher DPPH scavenging activity per gram seaweed (dry basis) as those samples had higher yield of dry lipid extract than UGB and PI samples. Fu et al. [30] found that the DPPH scavenging activities of Malaysian *S. polycystum* ethanolic extracts derived from different extraction parameters (solvent percentage, solid to solvent ratio, temperature, and time) were around 0.1–0.9 µmol Trolox equivalent (TE)/g dry weight of seaweed. Those were still lower than the DPPH scavenging activities observed in this study (0.96–1.58 µmol TE/g seaweed on a dry basis). Moreover, Ummat et al. [31] reported that the DPPH radical scavenging activities of the ethanolic extracts derived from 11 different brown seaweed species through conventional extraction process ranged from 9.98–82.70 µmol TE/g extract. They also found that ultrasound-assisted extraction could enhance the DPPH scavenging activities which reached 20.78–116.26 µmol TE/g extract. According to the reports of Budhiyanti et al. [32], different extracts (membrane-bound and cytoplasmic extracts) from *Sargassum* sp. (450 ppm) provided DPPH radical scavenging effects in the range of 0.17–48.71%, similar to data found in this work. Nevertheless, the antioxidant effects of the *Sargassum* sp. extracts were much lower than that of the synthetic BHT (butylated hydroxytoluene) since a lower concentration (100 ppm) provided a 90% DPPH scavenging effect. Synthetic antioxidants have been widely used because of their higher stability, performance, and wide availability, but their safety issues have been raised over timer. Synthetic antioxidants, including BHT and BHA, are reported to be responsible for several side effects such as carcinogenesis and liver damage [33,34]. In addition, the worldwide trend toward the usage of natural compounds encourages massive exploration of natural antioxidants as replacements for synthetic ones. In this regard, seaweed has been largely studied as one of the richest sources of natural antioxidants [34].

The differences in DPPH scavenging activities shown by various studies could be due to differences in species, extraction methods, and environmental conditions [30,31,32]. A study by Silva et al. [35] showed that ferric reducing antioxidant power of different lipid-soluble fractions derived from brown seaweed *Bifurcaria bifurcata* were ranging from 7.64–1128.20 µmol FeSO_4_/g fraction, while the FRAP values found in this study were about 634.88–969.31 µmol FeSO_4_/g lipid extract. Those observed values were still within the range found in the aforementioned study.

According to Pearson’s correlation analysis (Table 2), there is a positive correlation between antioxidant activity (DPPH and FRAP assay) and anti-inflammatory activity (both pre-incubated and co-incubated cell culture model). The antioxidant capacity of the bioactive compounds in *Sargassum* extract might contribute to suppressing oxidative stress status in the inflamed cells [36], so the positive reciprocal feedback loop between inflammation and oxidative stress could be interrupted. UGB was found to show the strongest NO-inhibition capacity in the two indicated models. Moreover, UGB also gave the highest ferric reducing antioxidant power which reached 969.31 ± 47.88 µmol FeSO_4_/g lipid extract, while its DPPH radical scavenging activity per tested lipid extract was not significantly different (*p* > 0.05) from PI and AB samples. It could indicate that the reducing power of the tested extract might greatly contribute to interfere with inflammatory response in LPS-induced RAW 264.7 cells.

The FRAP method is known as the most putative method which reflects the electron-donating capacity of the bioactive compounds or food components. The transferred electron will then reduce any compound, including metals, carbonyl groups, and radicals [37]. Although the ability of bioactive compounds to reduce iron has little relationship to the radical quenching processes (hydrogen atom transfer) mediated by most antioxidants, the oxidation or reduction of radicals into ions still stops radical chains. Furthermore, DPPH radical scavenging activity of bioactive compound delineates the electron-donating capacity as the main reaction and hydrogen-atom abstraction as a marginal reaction [38]. Rajauria [39] reported that the ferric reducing antioxidant power and DPPH radical scavenging activity of brown seaweeds (*Himanthalia elongata, Saccharina latissima-formerly Laminaria saccharina* and *Laminaria digitata*) lipophilic extracts were positively correlated with their total phenol, flavonoid, carotenoid, and chlorophyll contents. Syad et al. [40] found that the presence of a high amount of terpenoid compounds in non-polar extract of brown seaweed *S. wightii* could be the possible reason for its potential antioxidant activity, characterized by its strong DPPH, OH·, and H_2_O_2_ radical scavenging activity, and the high ferric reducing antioxidant power.

Several compounds that are potentially responsible for the anti-inflammatory activity of *Sargassum* non-polar fraction include fucosterol or other steroid compounds, fucoxanthin and its derivatives, fatty acids and simple organic compounds such as hexadecanoic acid, neophytadiene, tetradecanoic, 8-heptadecene, and 3,7,11,15-tetramethyl-2-hexadecen-1-ol, omega-3 fatty acids (C20:5n3 and C18:4n3), some phenolic compounds, and other pigment compounds contained in *Sargassum* [6]. According to our study results, UGB and AB samples exhibited the most promising anti-inflammatory and antioxidant effects. Although PI sample also showed a strong NO-inhibition effect on RAW 264.7 cells after LPS stimulation, this sample was not considered as potential anti-inflammatory sources because it gave the highest cytotoxicity to the cells. Both cytotoxic and biological activities of bioactive components need to be considered because they provide an overview of the safety and efficacy of the tested compound [41,42,43]. In the anti-inflammatory screening studies, Adebayo et al. [41] and Somchit et al. [42] used the selectivity index (SI) to ascertain that the observed anti-inflammatory effect of the tested compound was not due to a general metabolic toxic effect on the RAW 264.7 cell lines. A higher SI value is indicative of selective anti-inflammatory activity while a low SI value indicates higher cellular toxicity. SI is calculated by dividing the half-maximal cytotoxic concentration (CC50) by the half-maximal inhibitory concentration (IC_50_). When comparing between AB and UGB treatment (Appendix A), AB treatment was found to have higher selectivity index (3.19 in pre-incubated model and 2.03 in co-incubated model) than UGB (2.17 in pre-incubated model and 1.54 in co-incubated model).

The biological activity of tested samples will be strongly influenced by their bioactive content. Furthermore, the bioactive content of seaweed, especially lipid-soluble component, is affected by several factors, such as genetic, algal life stage, physiological status, light intensity, season, hydrodynamic condition, and other environmental parameters like temperature, pH, salinity, dissolved oxygen, total nitrogen, total phosphorus, etc [44,45]. Our previous preliminary study on chemical profiling of *S. ilicifolium* (as *S. cristaefolium*) from different coastal areas showed that all tested samples have distinct lipid-soluble profiles with different morphological characteristics [46]. According to the principal component analysis (PCA) results on lipid-soluble components in that study, SHB and AB samples were clustered together at close proximity, whilst PI and UGB samples are located in the opposite F1 score range (positive factor score). SHB and AB were characterized by larger blade size, higher content of chlorophyll, fucoxanthin, carotenoid, polyunsaturated fatty acids (PUFA), total n-3 fatty acids, total n-6 fatty acids, and also a lower ratio of n-6 to n-3. While PI and UGB were characterized by smaller blade size, higher content of saturated fatty acids/SFA (C14:0, C15:0, C16:0, C18:0, C20:0), some monounsaturated fatty acids/MUFA (C16:1, C17:1, C18:1n9, C22:1n9), C20:2, and C22:6n3. When comparing PI and UGB samples, UGB were found to have higher content of C12:0, C18:2n6, C18:3n6, C20:1n9, C20:5n3, and total PUFA than PI. The results of gas chromatography-flame ionization detector (GC-FID) analysis on samples’ FAME are attached in the Appendix A).

In the present study, we found that differences in water groups (southern or northern water of Java) did not appear to affect the tendency of anti-inflammatory and antioxidant activity. UGB and AB samples, which represented two distinctive water groups, exhibited a promising anti-inflammatory and antioxidant effects. Both of them have a distinct lipid-soluble profile as previously described. Based on morphological observation of fresh seaweeds, UGB had a narrow blade size (0.7 ± 0.2 cm width and 1.6 ± 0.3 cm length) with green color, while AB had a wide blade size (1.8 ± 0.2 cm width and 4.5 ± 0.3 length) with a dark-brown color [46]. The different morphological appearances might indicate different developmental stages [47]. However, the thallus size and morphological appearance can also be greatly influenced by the natural environmental condition [12,14]. Koivikko [48] stated that the degree of polymerization of phlorotannins, a typical phenolic compound in brown seaweed, will be different among developmental stages, young thallus tends to contain a high amount of short oligomers, while adult thallus will accumulate longer and more complex forms of phlorotannins, which are more difficult to degrade or exude. Actually, data regarding the content and composition of lipid-soluble phenolic compounds were not available in this study, but this group of compounds may also be responsible for the anti-inflammatory and antioxidant activity of the lipid-soluble/non-polar fraction of *Sargassum* as summarized by Saraswati et al. [6]. Although lipid class composition (neutral lipid, glycolipid, and phospholipid) of UGB and AB samples were not significantly different [46], their qualitative composition would strongly influence their bioactivity. Tasende [49] revealed that the distribution of sterols composition (part of neutral lipid fraction) was greatly affected by the algal life cycle and this would have an implication to the bioactivity of algal lipid.

Based on the facts found in this study, the single compound responsible for providing anti-inflammatory and antioxidant effects could not be well defined. According to the results of Pearson’s correlation analysis on lipid-soluble components from our previous study [46] and the bioactivities (anti-inflammatory and antioxidant) from our present study (Appendix A), the lipid-soluble components which were found to be positively correlated with anti-inflammatory activity (both models), DPPH scavenging activity, and ferric reducing antioxidant power in a consistent manner were only chlorophyll a, neutral lipid content, total SFA, and total MUFA content. However, this is not sufficient to draw conclusions, because the metabolite information contained in the crude lipid extract was not comprehensive enough. For example, the qualitative composition of neutral lipid which could influence the observed bioactivity was not available in this study.

A high-throughput approach using mass spectroscopy (MS) or nuclear magnetic resonance (NMR) methods may provide a clearer picture of the responsible compounds for bioactivity. The study of Saraswati et al. [50] reported that there were several putative compounds in the lipid-soluble fraction of S. *ilicoflium* (as *S. cristaefolium*) that had a strong correlation with NO inhibition activity in LPS-induced RAW 264.7, DPPH scavenging activity, and ferric reducing antioxidant power. These compounds included porphyrin derivatives (pheophytin a, 13^2^-hydroxypheophytin a, pheophorbide a), all-trans fucoxanthin, and some monogalactosyldiacylglycerols (MGDG), such as MGDG (16:0/18:1), MGDG (20:5/18:3), and MGDG (18:3/18:4). That study used a metabolomic approach to determine the responsible compounds for bioactivity. Metabolite profiling in that study was performed using ultra-high performance liquid chromatography-electrospray ionization orbitrap tandem mass spectrometry (UHPLC-MS/MS).

Although the results of the Pearson’s correlation analysis between omega-3 PUFA or fucoxanthin content and anti-inflammatory activity in this study did not show a positive association, the high content of omega-3 PUFA and fucoxanthin in AB samples might contribute to the reported anti-inflammatory activity according to the facts reported by Saraswati et al. [50]. Ahmad et al. [51] stated that a complex mixture of saturated, monounsaturated, and polyunsaturated fatty acids may all impact the anti-inflammatory activity of marine lipid extracts. Eventually, we can conclude that the observed anti-inflammatory and antioxidant effect in this study might be generated from the cumulative effect of all lipid-soluble constituents contained in the brown seaweed lipid extract.

## 3. Materials and Methods

### 3.1. Materials

Murine macrophage RAW 264.7 cells (TIB-71™, ATCC, Manassas, VA, USA) were used as a cell culture model for anti-inflammatory activity screening. Other materials used in this study included powdered Roswell Park Memorial Institute (RPMI) 1640 medium, 6-hydroxy-2,5,7,8-tetramethylchroman-2-carboxylic acid or Trolox, 2,2-diphenyl-1-picrylhydrazyl (DPPH), 2,4,6-tris(2-pyridyl)-s-triazine (TPTZ), lipopolysaccharides (LPS) of *Escherichia coli* O111:B4, sodium nitrite, and Griess reagent from Sigma UK (Gillingham, UK), penicillin-streptomycin and fetal bovine serum (FBS) (Gibco^TM^ Life Technologies Corporation, Gaithersbug, MD, USA), chloroform and methanol from Merck KGaA (Darmstadt, Germany), and other reagents used for anti-inflammatory and antioxidant activity assay.

### 3.2. Brown Seaweed Sampling and Sample Preparation

Brown seaweed samples were harvested from four different coastal areas, i.e., Awur Bay, Jepara, Central Java (6°36′54″ S, 110°38′55″ E) and Pari Island, Seribu Islands, DKI Jakarta (5°51′48″ S, 106°36′29″ E), Sayang Heulang Beach, Garut, West Java (7°40′12″ S, 107°41′50″ E) and Ujung Genteng Beach, Sukabumi, West Java (7°21′39″ S, 106°24′10″ E) in the same monsoon season (March–April 2017). Awur Bay and Pari Island are located in the northern waters of Java Island, while Sayang Heulang Beach and Ujung Genteng Beach are included as southern parts of Java Island. Seaweed identification was conducted at Marine Hydrobiology Division, Department of Marine Science and Technology, Bogor Agricultural University.

Fresh seaweed was firstly cleaned from physical impurities (e.g., sand and gravel) and dried before the extraction step. Drying was carried out using an oven dryer at 50 °C until seaweed moisture content reached <10%. The dried seaweed was ground to yield seaweed powder (18 mesh particle size). Dried seaweed powder was then stored at 4 °C.

### 3.3. Preparation of Crude Lipid Extract and The Chemical Characterization

A crude lipid extract was prepared by referring to the method of Bligh and Dyer [52] with modification as described by Susanto et al. [53]. The modification applied was rehydration of dried seaweed using distilled water (1:9, *w*/*v*) for one hour. Maceration of rehydrated seaweed was performed at room temperature for two hours using the mixture of chloroform (C) and methanol (M) at a ratio of 1:2 (*v*/*v*). A lipid extract was subsequently filtered by Buchner funnel with Whatman filter paper No. 1. The resulted filtrate was added using chloroform and distilled water to get the final ratio of C/M/W at 1:1:0.9. The chloroform layer containing crude lipid extract was separated and the solvent was evaporated using a rotary vacuum evaporator (R-300, Büchi, Flawil, Switzerland) with water bath temperature and vacuum pressure set to 50 °C and 332 bar, respectively. The remaining solvent in crude lipid extract was then evaporated by nitrogen flushing. The dried crude lipid extract was stored at −20 °C until further analysis. Crude lipid extraction procedure was performed in triplicate. Chemical characterization of crude lipid extract had been performed in our previous study [46]. The observed chemical characteristic parameters included total lipid, pigment profile (chlorophyll a, chlorophyll c, fucoxanthin, β-carotene), lipid class composition (neutral lipid, glycolipid, phospholipid), and fatty acid profile.

### 3.4. Bioactivity Examination

#### 3.4.1. Cell Culture and Cell Viability Assay

Murine macrophage RAW 264.7 cells were grown in the RPMI 1640 medium with inactivated FBS 10%, NaHCO_3_ 2%, and penicillin-streptomycin 100 U/mL in a humidified incubator (CO_2_ 5%, 37 °C). The cells were grown to 80–90% confluence, harvested by scraping, and diluted in fresh medium to the desired concentration of cells per ml. Cell viability was determined by methylthiazolyl tetrazolium (MTT) reduction assay [54]. The medium containing RAW 264.7 cells was cultured in a 96-well plate at a density of 10^5^ cells per mL. The plate was incubated overnight and then treated by 100 µL medium containing crude lipid extract at different concentrations (12.5–1000 µg/mL). After 24 h of incubation and cells rinsing by phosphate buffer saline (PBS), about 50 µg MTT reagent was added to each well and the plate was incubated for another 4 h at 37 °C. Formazan crystal was dissolved by 100 µL ethanol 96%. Optical density was measured at 595 nm using a microplate reader (BioRad, Hercules, CA, USA). The optical density of formazan formed by untreated cells (control) was taken as 100% viability.

#### 3.4.2. Anti-Inflammatory Activity Assay

Anti-inflammatory activity of brown seaweed crude lipid extract was observed through nitric oxide (NO) inhibition level in lipopolysaccharide (LPS)-induced RAW 264.7 cells. Cells were firstly seeded at a density of 10^4^ cells/well in a 96-microwell plate overnight. After cell adherence, cells were treated by crude lipid extracts and LPS under two different models, namely pre-incubated and co-incubated cell culture models as previously described by Wen et al. [55]. The doses of samples given to the cells were 12.5, 25, and 50 µg/mL. In pre-incubated cell culture model, cells were firstly treated by crude lipid extract at different concentrations for 24 h and followed by LPS stimulation (1 µg/mL) for another 24 h. Before LPS stimulation, the culture medium was discarded to remove samples and cells were rinsed by phosphate buffer saline thrice. In co-incubated cell culture model, cells were treated by crude lipid extract and LPS (1 µg/mL) simultaneously for 24 h in a humidified atmosphere (CO_2_ 5%, 37 °C). Cell culture supernatant was harvested and stored at −80 °C until further analysis. Anti-inflammatory assay was performed in triplicate. Medium in the anti-inflammatory assay was similar to the medium used in the cell culture preparation, i.e., RPMI 1640 with inactivated FBS 10%, NaHCO_3_ 2%, and penicillin-streptomycin 100 U/mL.

NO detection was performed according to the study of Rao et al. [56]. The amount of accumulated nitrite was used as an indicator of NO production in the culture medium. About 50 μL cell culture medium was mixed with 50 μL of the Griess reagent. Subsequently, the mixture was incubated at room temperature for 15 min in the dim light. The absorbance was measured at 540 nm using a UV-Vis microplate spectrophotometer (BioTek, Winooski, VT, USA). The standard calibration curve was prepared using sodium nitrite solution (1.56 to 100 mM). Fresh culture medium was used as a blank in every experiment.

#### 3.4.3. 2,2-Diphenyl-1-Picrylhydrazyl (DPPH) Scavenging Assay

The seaweed extract in methanol (1 mL, 500 ppm) was mixed with 2 mL of 0.08 mM methanolic solution of DPPH in a test tube [32]. Then, the mixture was vortexed and left for 30 min at room temperature in the dark. The absorbance was measured at 517 nm using a UV-Vis spectrophotometer. A calibration curve was made using Trolox as a standard. Antioxidant activity was expressed as µmol Trolox equivalent (TE)/g fraction. A blank was made by mixing 1 mL of methanol with 2 mL of 0.08 mM methanolic solution of DPPH. DPPH radical scavenging effect (RSE) was calculated using the following formula:RSE (%) = 1 − (Absorbance of sample/Absorbance of blank) × 100%.

#### 3.4.4. Ferric Reducing Antioxidant Power (FRAP) Assay

FRAP reagent was prepared from 2.5 mL of 10 mM 2,4,6-tris(2-pyridyl)-s-triazine (TPTZ) solution in 40 mM hydrochloric acid with 2.5 mL of 20 mM iron (III) chloride and 25 mL of 300 mM acetate buffers at pH 3.6. The FRAP reagent was prepared fresh daily and warmed to 37 °C in a water bath. 200 µL of sample extract (500 ppm) was added to 1.3 mL of FRAP reagent and was allowed to react for 30 min in a 37 °C water bath. The absorbance of the reaction mixture was recorded at 593 nm using a UV-Vis spectrophotometer. The standard curve was constructed using ferrous sulfate and the result was expressed in µmol ferrous equivalent (FE)/g fraction [57].

### 3.5. Statistical Analysis

Data are presented as mean ± standard deviation (SD). The effects of different samples to observed parameters were analyzed by one-way analysis of variance (ANOVA) and followed by Duncan posthoc test by IBM SPSS 20 (IBM, North Castle, NY, USA).

## 4. Conclusions

Crude lipid extracts of UGB and AB samples gave a promising anti-inflammatory effect in the in vitro model of LPS-induced RAW 264.7 cells. Intracellular action is thought to greatly contribute to the anti-inflammatory effect observed in this study, as the stronger NO-inhibition effect was found in the pre-incubated cell culture model than in co-incubated model. AB treatment showed a promising NO inhibition activity with a more acceptable cytotoxic effect at the indicated dose range. The anti-inflammatory activity of brown seaweed lipid extracts was positively correlated with their antioxidant activity (both DPPH and FRAP). These reported biological activities can be the basis for further development of *S. ilicifolium* as a source of functional ingredient.

## Figures and Tables

**Figure 1 marinedrugs-19-00252-f001:**
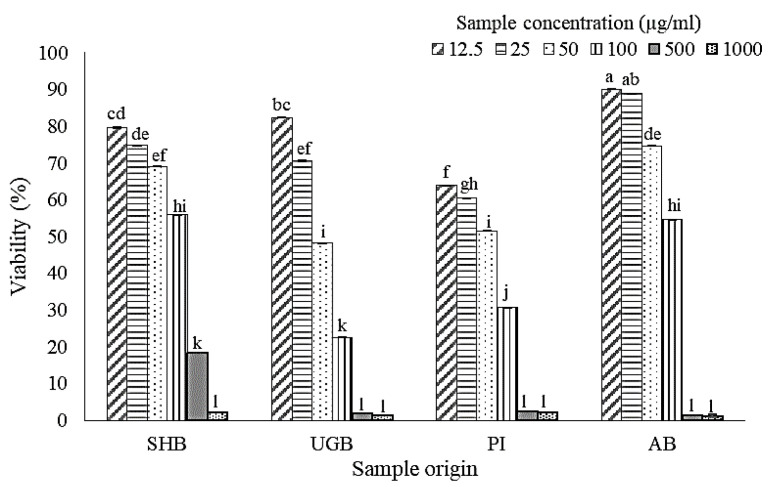
Viability of RAW 264.7 cells after incubated by *Sargassum ilicifolium* crude lipid extracts (12.5–1000 µg/mL) for 24 h. Note: SHB = Sayang Heulang Beach, UGB = Ujung Genteng Beach, PI = Pari Island, AB = Awur Bay. Means with at least one same letter are not significantly different based on Duncan’s test (*p* > 0.05). Error bars indicate the value of standard deviation.

**Figure 2 marinedrugs-19-00252-f002:**
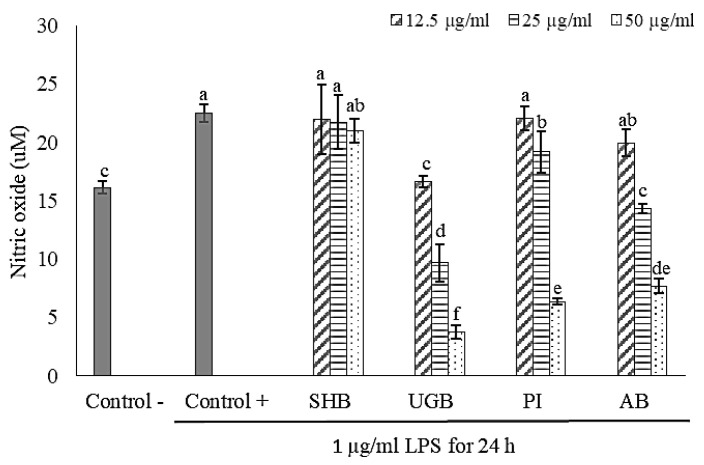
Effect of *Sargassum ilicifolium* crude lipid extracts (12.5–50 µg/mL) on LPS-induced NO production in pre-incubated cell culture model. RAW 264.7 cells (10^4^ cells/well in 96-well culture plates) were incubated by sample for 24 h before LPS stimulation. Note: SHB = Sayang Heulang Beach, UGB = Ujung Genteng Beach, PI = Pari Island, AB = Awur Bay. Means with at least one same letter are not significantly different based on Duncan’s test (*p* > 0.05). Error bars indicate the value of standard deviation.

**Figure 3 marinedrugs-19-00252-f003:**
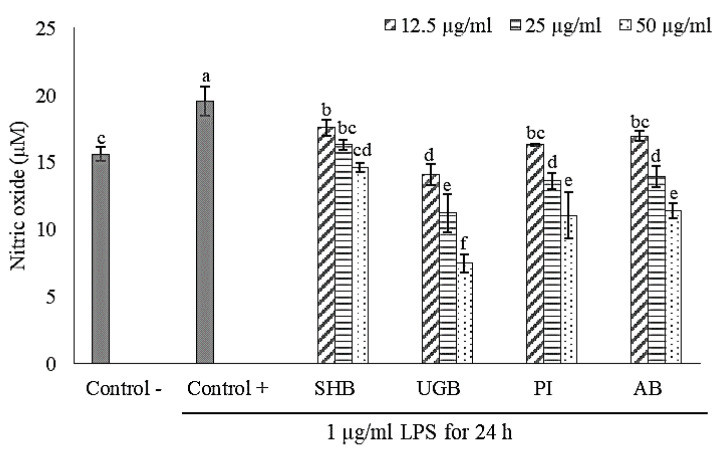
Effect of *Sargassum ilicifolium* crude lipid extracts (12.5–50 µg/mL) on LPS-induced NO production in co-incubated cell culture model. RAW 264.7 cells (10^4^ cells/well in 96-well culture plates) were incubated by sample and LPS for 24 h simultaneously. Note: SHB = Sayang Heulang Beach, UGB = Ujung Genteng Beach, PI = Pari Island, AB = Awur Bay. Means with at least one same letter are not significantly different based on Duncan’s test (*p* > 0.05). Error bars indicate the value of standard deviation.

**Table 1 marinedrugs-19-00252-t001:** Total lipid and antioxidant activity of *Sargassum ilicifolium* from different coastal areas.

Observed Parameters	Sample Origin			
	SHB	UGB	PI	AB
Total lipid (g/100 g seaweed db) ^1^	3.55 ± 0.29 ^b 2^	2.76 ± 0.28 ^a^	2.73 ± 0.19 ^a^	4.32 ± 0.09 ^c^
Antioxidant activity (µmol TE/g seaweed db) ^1^	1.27 ± 0.01 ^c^	1.01 ± 0.00 ^b^	0.96 ± 0.03 ^a^	1.58 ± 0.001 ^d^
Antioxidant activity (µmol TE/g lipid extract) ^1^	34.61 ± 0.17 ^a^	37.87 ± 0.08 ^b^	37.24 ± 1.15 ^b^	36.93 ± 0.18 ^b^
DPPH scavenging effect (%) with extract concentration of 500 ppm ^1^	39.6 ± 0.15 ^a^	42.57 ± 0.08 ^b^	42.00 ± 1.05 ^b^	41.71 ± 0.17 ^b^
FRAP (µmol FeSO_4_/g seaweed db) ^1^	23.44 ± 0.41 ^d^	25.98 ± 1.28 ^b^	16.32 ± 0.04 ^c^	29.24 ± 0.32 ^a^
FRAP (µmol FeSO_4_/g lipid extract) ^1^	637.06 ± 11.12 ^a^	969.31 ± 47.88 ^c^	634.88 ± 1.50 ^a^	681.58 ± 7.48 ^b^

^1^ All observed parameters values are given as mean ± SD. ^2^ Means in the same row with different superscripts differ significantly by Duncan’s test (*p* < 0.05). Note: SHB = Sayang Heulang Beach, UGB = Ujung Genteng Beach, PI = Pari Island, AB = Awur Bay, db = dry basis.

**Table 2 marinedrugs-19-00252-t002:** Relationship between antioxidant and anti-inflammatory activity of crude lipid extract of *S. ilicifolium.*

Treatment	Pearson’s Correlation Coefficient
DPPH vs. FRAP	0.484
DPPH vs. NO Inhibition in Pre-incubated Model	0.758 **
FRAP vs. NO Inhibition in Pre-incubated Model	0.749 **
DPPH vs. NO Inhibition in Co-incubated Model	0.794 **
FRAP vs. NO Inhibition in Co-incubated Model	0.819 **
NO Inhibition in Pre-incubated Model vs NO Inhibition in Co-incubated Model	0.865 **

Note: Data were statistically analyzed using Pearson’s correlation coefficient test. ** indicate correlation is significant at 0.01 level (2-tailed). DPPH means 2,2-diphenyl-1-picrylhydrazyl radical scavenging activity of crude lipid extract, while FRAP means ferric reducing antioxidant power.

## Data Availability

Not applicable.

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
