# Peer review of "Screening of In-Vitro Anti-Inflammatory and Antioxidant Activity of Sargassum ilicifolium Crude Lipid Extracts from Different Coastal Areas in Indonesia"

_marinedrugs, 2021, doi:10.3390/md19050252_

Round 1

Reviewer 1 Report

Dear authors

Thank you for your article. I find the article well constructed, precise with interesting results.

The only negative point is that you only talk about extracts. I think it would be judicious to detail the composition of the fractions by referring to publication number 38. These results would enrich the work. You could also relate the structure to the activity. 

Some remarks:

**line 365 what is TPTZ?
**standards are missing in table 1 and for antioxidant tests.

** Don't forget to underline all Latin names and also in the references.
**Finally, reference 1 is in Indonesia, I encourage you to prefer an English language publication. Check the FAO website, the latest reports on algae are very interesting and should be mentioned

Good luck

Best regards

Author Response

Comment:

The only negative point is that you only talk about extracts. I think it would be judicious to detail the composition of the fractions by referring to publication number 38. These results would enrich the work. You could also relate the structure to the activity. 

Response:

Thank you very much for your critical view of our manuscript. First of all, we acknowledge the limitations of this screening study. Metabolite information was not available in a comprehensive manner, so the correlation between bioactivity and structure may be biased. However, we tried to analyze the relationship between the lipid-soluble components in the tested extracts and their bioactivities (Table S2). The possible bias of the analysis was overcome by citing studies related to lipid-soluble components which are responsible for anti-inflammatory and antioxidant activity, one of which is our previously published study (Ref No. 48) which also used the same species but the metabolite profiling was performed using a high-throughput approach. The major revisions in terms of discussion substance have been marked with track changes. 

Comment:

line 365 what is TPTZ?

Response:

We have defined that term in method section.

Comment:

**standards are missing in table 1 and for antioxidant tests.

Response:

Yes, we did not use the positive control in this case. But we tried to express the antioxidant activity results using standard (Trolox and FeSO4 equivalent per gram sample or dry lipid extract). Therefore, we can compare our results with other published studies and see how our data are. The revision of the discussion of section 2.3 is shown on page 10. As the main objective of this study, the resulting trend of antioxidant activity data will serve as an illustration for future utilization.

Comment:

** Don't forget to underline all Latin names and also in the references.

Response:

We have corrected that issue.

Comment:

**Finally, reference 1 is in Indonesia, I encourage you to prefer an English language publication. Check the FAO website, the latest reports on algae are very interesting and should be mentioned

Response:

Thanks a lot for your valuable input. We have revised reference 1, we used the data of the Ministry of Marine and Fisheries Republic of Indonesia which were presented in both English and Indonesian language. In addition, the latest reports by FAO was also included as reference 2.

Reviewer 2 Report

The manuscript is a continuation of previously published paper. It is clearly written and the topic is interesting. However, I have some questions/suggestions before acceptance for publication. The specific list is below:

Line 24: Use shortcut: S. cristaefolium , the same in line 67 … (check all text)

Line 44: Rewrite the phrase: „responsible inflammatory agents” – responsible for inflammatory activity?

2.1. Section: Any control, e.g. sample without lipid extract?

Fig.1-3 error bars mean standard deviation or standard error? It should be pointed out in desription under the Figures.

Table 1. Explain the shortcut „db” under the Table

Line 244-246: it is not clear whether Authors cited the results from previous study. TPC was not assessed in this manuscript. Moreover, it is unusual that total phenolics content (hydrophilic components) is investigated in lipidic fraction

- Lack of italic for the Latin name of the plant in some places e.g. in line 76 (check all text)

- Line 269-270 is unnecessary because these information are in line 281.

- Line 297: Why such modification was applied? It should be clarified.

Author Response

Comment:

The manuscript is a continuation of previously published paper. It is clearly written and the topic is interesting. However, I have some questions/suggestions before acceptance for publication. The specific list is below:

Line 24: Use shortcut: S. cristaefolium , the same in line 67 … (check all text)

Response:

Thank you very much for your valuable input for our manuscript. We have corrected that issue. The species name was revised to S. ilicifolium as recommended by reviewer 3.

Comment:

Line 44: Rewrite the phrase: „responsible inflammatory agents” – responsible for inflammatory activity?

Response:

We have revised that phrase as you recommend

Comment:

2.1. Section: Any control, e.g. sample without lipid extract?

Response:

As mentioned in method section 3.4.1, the optical density of formazan formed by untreated cells (control) was taken as 100% viability. So, the viability values were relative to the control (cells without lipid extract).

Comment:

Fig.1-3 error bars mean standard deviation or standard error? It should be pointed out in description under the Figures.

Response:

We have made the revisions in the caption of Figure 1-3.

Comment:

Table 1. Explain the shortcut „db” under the Table

Response:

We have defined the term of “db” under the table.

Comment:

Line 244-246: it is not clear whether Authors cited the results from previous study. TPC was not assessed in this manuscript. Moreover, it is unusual that total phenolics content (hydrophilic components) is investigated in lipidic fraction

Response:

Actually, we have conducted the TPC analysis for all crude lipid extract by referring to the method of Chandini et al. (2008), because our previous summary (Ref 6) stated that phenolic compound could be responsible for the anti-inflammatory activity of Sargassum non-polar fraction. But because those data did not support our results currently, so we have deleted those parts.

Comment:

- Lack of italic for the Latin name of the plant in some places e.g. in line 76 (check all text)

Response:

We have re-checked it and made corrections to that issue.

Comment:

- Line 269-270 is unnecessary because these information are in line 281.

Response:

Thank you very much for your correction. We have solved that problem.

Comment:

- Line 297: Why such modification was applied? It should be clarified.

Response:

The modification applied was rehydration of dried seaweed using distilled water (1:9, w/v) for one hour. That explanation has been placed in the next sentence after Susanto et al.

Reviewer 3 Report

marinedrugs-1175723-peer-review-v1-rev

The manuscript entitled "Screening of In-vitro Anti-Inflammatory and Antioxidant Activity of Sargassum cristaefolium Crude Lipid Extracts from Different Coastal Areas in Indonesia" addresses the assessment of the antioxidant and anti-inflammatory capacity of a brown macroalgae present in several zomas in Indonesia. The manuscript is well structured and reasonably well written. However, the authors will have to make an in-depth review at the taxonomy level, since the used names are synonymous and not the valid names nowadays. 

Fo example, the correct (taxonomic valid) name is Sargassum ilicifolium

Corrections needed:

line 3 - Sargassum ilicifolium

line 15 - Sargassum ilicifolium

line 24 - Sargassum ilicifolium

line 26 - Sargassum ilicifolium

line 34 -  Kappaphycus alvarezii (formerly Eucheuma cottonii)

line 35 - Eucheuma denticulatum (formerly E. spinosum)

line 44 - Sargassum (in italics)

line 51 - Sargassum ilicifolium (formerly Sargassum cristaefolium)

line 57/58 - In addition, S. duplicatum, S. berberifolium, S. cristaefolium and Fucus latifolius are recognized as synonym of Sargassum ilicifoliumm, according Guiry and Guiry [11].

Note: remove the reference "Mattio, L.; Payri, C.E.; Verlaque, M. Taxonomic revision and geographic distribution of the subgenus Sargassum (FUCALES, PHAEOPHYCEAE) in the western and central pacific islands based on morphological and molecular analyses. J. Phycol. 2009
45, 1213–1227." and add this new reference "Guiry, M.D. & Guiry, G.M. 2021. AlgaeBase. World-wide electronic publication, National University of Ireland, Galway. https://www.algaebase.org/search/species/detail/?species_id=4580; searched on 31 March 2021."

line 59/60 - Sargassum ilicifolium (as Sargassum cristaefolium)

line 67 - Sargassum ilicifolium

line 76 - S. ilicifolium (in italics)

line 79 - S. ilicifolium

line 92 - 69.01 and 74.64%, respectively.

line 97 - Sargassum ilicifolium

line 101 - S. ilicifolium

line 134 - Sargassum ilicifolium

line 140 - Sargassum ilicifolium

line 145 - S. ilicifolium

line 151 - Padina australis (in italics)

line 161 - S. ilicifolium

line 180 - S. ilicifolium

line 195 - S. ilicifolium

line 207 - of brown seaweeds (Himanthalia elongata, Saccharina latissima - formerly Laminaria saccharina, and Laminaria digitata

line 225 - S. ilicifolium

line 269 - Sargassum ilicifolium

line 340 - ... simultaneously for 24 h in ...

line 341 - midified atmosphere (CO2 5%, 37 °C).

line 344 - ..., NaHCO3 2%, and ...

line 380 - in vitro (in italics)

line 387 - S. ilicifolium

Author Response

Comment:

The manuscript entitled "Screening of In-vitro Anti-Inflammatory and Antioxidant Activity of Sargassum cristaefolium Crude Lipid Extracts from Different Coastal Areas in Indonesia" addresses the assessment of the antioxidant and anti-inflammatory capacity of a brown macroalgae present in several zomas in Indonesia. The manuscript is well structured and reasonably well written. However, the authors will have to make an in-depth review at the taxonomy level, since the used names are synonymous and not the valid names nowadays. 

For example, the correct (taxonomic valid) name is Sargassum ilicifolium

Corrections needed:

line 3 - Sargassum ilicifolium

line 15 - Sargassum ilicifolium

line 24 - Sargassum ilicifolium

line 26 - Sargassum ilicifolium

line 34 -  Kappaphycus alvarezii (formerly Eucheuma cottonii)

line 35 - Eucheuma denticulatum (formerly E. spinosum)

line 44 - Sargassum (in italics)

line 51 - Sargassum ilicifolium (formerly Sargassum cristaefolium)

line 57/58 - In addition, S. duplicatum, S. berberifolium, S. cristaefolium and Fucus latifolius are recognized as synonym of Sargassum ilicifoliumm, according Guiry and Guiry [11].

Note: remove the reference "Mattio, L.; Payri, C.E.; Verlaque, M. Taxonomic revision and geographic distribution of the subgenus Sargassum (FUCALES, PHAEOPHYCEAE) in the western and central pacific islands based on morphological and molecular analyses. J. Phycol. 2009
45, 1213–1227." and add this new reference "Guiry, M.D. & Guiry, G.M. 2021. AlgaeBase. World-wide electronic publication, National University of Ireland, Galway. https://www.algaebase.org/search/species/detail/?species_id=4580; searched on 31 March 2021."

line 59/60 - Sargassum ilicifolium (as Sargassum cristaefolium)

line 67 - Sargassum ilicifolium

line 76 - S. ilicifolium (in italics)

line 79 - S. ilicifolium

line 92 - 69.01 and 74.64%, respectively.

line 97 - Sargassum ilicifolium

line 101 - S. ilicifolium

line 134 - Sargassum ilicifolium

line 140 - Sargassum ilicifolium

line 145 - S. ilicifolium

line 151 - Padina australis (in italics)

line 161 - S. ilicifolium

line 180 - S. ilicifolium

line 195 - S. ilicifolium

line 207 - of brown seaweeds (Himanthalia elongataSaccharina latissima - formerly Laminaria saccharina, and Laminaria digitata

line 225 - S. ilicifolium

line 269 - Sargassum ilicifolium

line 340 - ... simultaneously for 24 h in ...

line 341 - midified atmosphere (CO2 5%, 37 °C).

line 344 - ..., NaHCO3 2%, and ...

line 380 - in vitro (in italics)

line 387 - S. ilicifolium

Response:

Thank you very much for your valuable input to our manuscript. We really appreciate your recommendation. We have checked the reference of Guiry and Guiry (2021) and decided to correct the taxonomy issue. We also have made revisions point-by-point as you reviewed. The major revisions in terms of discussion substance have been marked with track changes. 

Reviewer 4 Report

The MS 'Screening of In-vitro Anti-Inflammatory and Antioxidant Activity of Sargassum cristaefolium Crude Lipid Extracts from Different Coastal Areas in Indonesia' present the anti-inflammatory and antioxidant activities of four crude lipid extracts (previously characterized) obtained from Sargassum cristaefolium collected from different Indonesian areas. Authors show results of four batches of experiments using these extracts: cytotoxicity, antioxidant activity (DPPH and FRAP assays) and anti-infalmmatory activity (NO inhibition).

From my point of view, data provided is not enough to demonstrate the anti-inflammatory and antioxidant activities of the crude lipid extracts of Sargassum cristaefolium. Both material and methods section and results (and discussion) sections are very slightly described.

I find the presentation of the statistical results confusing. For example, in the Figure 1 it is relevant to understand which concentration of each extract reduces the cell viability against other concentrations of the same extract. It is not relevant to compare the cytotoxicity effect of different extracts at same concentration. Each extract has been demonstrated to have different lipid profile (although in terms of FAMEs, the chemical profile of UGB and AB samples seems to be more similar among them than when compared against SHB and PI, at least, according to figures showed in the supplementary material), and thus they can differently affect to cell survival.

Experiments developed for evaluating antioxidant capacity using both DPPH and FRAP assays were performed using just one concentration of algal crude lipids, 500 ppm, which in addition had been previously indicated as cytotoxic. Data obtained from these experiments is scarcely showed. Authors do not show any curve using different sample concentrations either show data/results of calibration curves. Besides, the section of results shows few mismatches and data is very scarcely compared with previously published works.

A small discussion has been introduced in results section following the short explanation of data but I find it very general and not conclusive.

I would rather suggest to authors to improve the presentation of their results in order to do not hinder the comparison with other previously published works and they can probably use the differential FAME profile to try to find some stronger correlation between these lipophilic profiles and the results they observed in terms of antioxidant and anti-inflammatory activities.

Some data, typo and English style errors I would like to underline. However, full MS need to be fully and carefully reviewed in order to solve many other mistakes:

P1, L20: The term UGB seems to refer to UG, please unify nomenclature

P2, L58 and P8, L335: Use italics for et al., please. Review full MS

P2, L76 and P4, L154: Use italics for species name, please. Review full MS

P3, L112: Change verb “are” for “is” otherwise use the word “effect” in plural

P5, L168-169: Review full sentence. Review and improve your results discussion, please

P7, L244-247: The total phenolic content of the samples UGB and PI are nearly same. Therefore, this data does not support your results, at least, as authors present them currently.

P8, L222: Replace “in the room temperature” for “at room temperature”

P8, L302: Replace “by” in this sentence “was added by chloroform” with “using”, for example

P9, L352: Check the spelling of units, please. I guess “Mm” means mM

Author Response

Comment:

The MS 'Screening of In-vitro Anti-Inflammatory and Antioxidant Activity of Sargassum cristaefolium Crude Lipid Extracts from Different Coastal Areas in Indonesia' presents the anti-inflammatory and antioxidant activities of four crude lipid extracts (previously characterized) obtained from Sargassum cristaefolium collected from different Indonesian areas. Authors show results of four batches of experiments using these extracts: cytotoxicity, antioxidant activity (DPPH and FRAP assays) and anti-inflammatory activity (NO inhibition).

From my point of view, data provided is not enough to demonstrate the anti-inflammatory and antioxidant activities of the crude lipid extracts of Sargassum cristaefolium. Both material and methods section and results (and discussion) sections are very slightly described. I find the presentation of the statistical results confusing. For example, in the Figure 1 it is relevant to understand which concentration of each extract reduces the cell viability against other concentrations of the same extract. It is not relevant to compare the cytotoxicity effect of different extracts at same concentration. Each extract has been demonstrated to have different lipid profile (although in terms of FAMEs, the chemical profile of UGB and AB samples seems to be more similar among them than when compared against SHB and PI, at least, according to figures showed in the supplementary material), and thus they can differently affect to cell survival.

Response:

Thank you very much for your in-depth review of our manuscript. We really appreciate your valuable input. First of all, we would like to clarify the statistical results. Actually, we use one-way ANOVA followed by Duncan’s posthoc test to see the difference amongst all mean values. In this case, different doses of extracts were not defined as a separated factor. Means with at least one same letter are not significantly different based on Duncan’s test (p>0.05).  

Second, we also would like to clarify the purpose of our study. Actually, the aim of this screening study was to see the potential of S. cristaefolium seaweed (now revised to S. ilicifolium according to the recommendation of Reviewer 3) from different coastal areas in Indonesia as a source of anti-inflammatory agents and antioxidants. In addition, this study served as a bridge between our sequential studies that ultimately determine which sample (from one origin) should be selected for further fractionation in our previously published manuscript in Food Research International (Ref. 48).

Regarding the cell viability assay, we conducted that step to determine doses of samples for anti-inflammatory activity assay. Determination of cell viability becomes important, so the observed anti-inflammatory effect is deemed not to be attributable to cytotoxic effects. In addition, the cell viability assay was also used to provide an overview of the selective NO inhibitory effect of potential candidates. To facilitate the evaluation of anti-inflammatory activity, the determination of the same dosage range is important because, in the end, the dose will determine the safety and effectiveness of the tested extracts. The results showed the same trend that increased dosage can increase the cytotoxicity effect. In all sample treatments, we could find that a dose of 100 µg/ml was unacceptable for anti-inflammatory activity assay because of its significant cytotoxicity effect.

In addition, the trend regarding the FAME profile had been actually described in the discussion section. UGB and AB samples seem to be more similar in the GC-FID chromatogram results because those samples were analyzed using same concentration of the internal standard (heptadecanoic acid), while SHB and PI samples were analyzed using 10-times more concentrated internal standard (we conducted the analysis in different batch, the note regarding this issue had been added in the supplementary materials) than in UGB and AB. But after the calculation of fatty acid content by considering the response factor of each fatty acid, we found that all samples had similar fatty acid distribution although they had marked morphological variations. The observed predominant fatty acids were palmitic, palmitoleic, oleic, linoleic, linolenic, arachidonic/ARA, and eicosapentaenoic acid/EPA. The highest ratio of omega-6 to omega-3 PUFA was found in PI sample (2.34), while the values of other samples ranged from 1.46-1.97.

Comment:

Experiments developed for evaluating antioxidant capacity using both DPPH and FRAP assays were performed using just one concentration of algal crude lipids, 500 ppm, which in addition had been previously indicated as cytotoxic. Data obtained from these experiments is scarcely showed. Authors do not show any curve using different sample concentrations either show data/results of calibration curves. Besides, the section of results shows few mismatches and data is very scarcely compared with previously published works.

Response:

Yes, we only use one concentration in this case. But we tried to express the antioxidant activity results using standard (Trolox and FeSO4 equivalent per gram sample or dry lipid extract). Standard/calibration curves (with R2>0.9) were of course created first before sample analysis. Therefore, we can compare our results with other published studies and see how our data are. The revision of the discussion of section 2.3 is shown on page 10. As the main objective of this study, the resulting trend of antioxidant activity data will serve as an illustration for future utilization. The use of different concentrations is usually intended to determine the IC50 value, but we did not do this in this study. So we used publications that used the same units for comparison of antioxidant activity (Fu et al. [30], Ummat et al. [31], Budhiyanti et al. [32], and Silva et al. [33]). Based on the results of comparisons with various publications, the antioxidant activity values ​​that we found were still within the range of values ​​that had been previously reported.

The reason for using the concentration at 500 ppm for determining the antioxidant activity was because this concentration provided an optimum absorbance value without pigment interference. The use of lower concentrations would provide less optimum absorbance values ​​(>0.2). At least, through testing the antioxidant activity, we can see a trend of the extract's capacity to scavenge DPPH radicals and reduce ferric ions which may contribute to their anti-inflammatory activity in a cell culture model. As a reaffirmation, we also present the value of antioxidant activity in the form of a standard equivalent.

Comment:

A small discussion has been introduced in results section following the short explanation of data but I find it very general and not conclusive. I would rather suggest to authors to improve the presentation of their results in order to do not hinder the comparison with other previously published works and they can probably use the differential FAME profile to try to find some stronger correlation between these lipophilic profiles and the results they observed in terms of antioxidant and anti-inflammatory activities.

Response:

First of all, we acknowledge the limitations of this screening study. Metabolite information was not available in a comprehensive manner, so the correlation between bioactivity and structure may be biased. However, we tried to analyze the relationship between the lipid-soluble components in the tested extracts and their bioactivities using Pearson’s correlation analysis (Table S2). The possible bias of the analysis was overcome by citing studies related to lipid-soluble components which are responsible for anti-inflammatory and antioxidant activity, one of which is our previously published study (Ref No. 48) which also used the same species but the metabolite profiling was performed using a high-throughput approach. The major revisions in terms of discussion substance have been marked with track changes (primarily section 2.3).

Comment:

Some data, typo and English style errors I would like to underline. However, full MS need to be fully and carefully reviewed in order to solve many other mistakes:

P1, L20: The term UGB seems to refer to UG, please unify nomenclature

P2, L58 and P8, L335: Use italics for et al., please. Review full MS

P2, L76 and P4, L154: Use italics for species name, please. Review full MS

P3, L112: Change verb “are” for “is” otherwise use the word “effect” in plural

P5, L168-169: Review full sentence. Review and improve your results discussion, please

P7, L244-247: The total phenolic content of the samples UGB and PI are nearly same. Therefore, this data does not support your results, at least, as authors present them currently.

P8, L222: Replace “in the room temperature” for “at room temperature”

P8, L302: Replace “by” in this sentence “was added by chloroform” with “using”, for example

P9, L352: Check the spelling of units, please. I guess “Mm” means Mm

Response:

We have revised the issues according to your review from point to point. Revisions are indicated by the track changes feature. Again, we are really thankful for your comprehensive input.

Round 2

Reviewer 4 Report

Dear authors,
I see you have work to try to improve your results. Thank you for all your effort, I really appreciate the effort you made to improve the discussion section including the Pearson study to try to correlate compositional and biological data.

Just few details I would like to underline:

L21: remove the adjective "strong" from the abstract

L190-194: Data collected from Budhiyanti et al.'s paper is presented in a confusing way. For instance:

Original paper says: "The presented data in Fig. 2 indicated that RSA in extract concentration of 0.45 mg mL−1 were in the range of 14.61-48.71% for membrane bound and 0.17-44.05% for cytoplasmic extracts. Nevertheless, the results of these extracts had lower antioxidant activity than the commercial antioxidant BHT (100 ppm, 89.83%)."

Authors claimed in this MS: "According to the reports of Budhiyanti et al. [32], the crude methanolic extracts derived from different Sargassum species at a concentration of 450 ppm could give the DPPH radical scavenging effects around 0.17-44.05%, lower than the commercial BHT (butylated hydroxytoluene) which gave 89.90% DPPH scavenging effect at a concentration of 100 ppm"

I believe that authors selected the report of Budhiyanti et al. because of the similarity of the data regarding the DPPH radical scavenging effect with authors' data.
First of all, from my point of view, authors should indicate the full range of DPPH radical scavenging effect (0.17 to 48.71%)  instead of providing the range relative to the cytoplasmic extract, at least, authors can demonstrate that their extracts act at cytoplasmic level.
Secondly, in my opinion the way the text has been redacted does not the idea of the low antioxidant activity of the extracts when compared against a synthetic antioxidant.
I would rewrite this sentence as follows:
"According to the report of Budhiyanti et al. [32], different crude methanolic extracts from Sargassum sp. (450 ppm) provided DPPH radical scavenging effects in the range of 0.17-48.71%, similar to data found in this work. Nevertheless, the antioxidant effects of the Sargassum sp. extract were much lower than that of the commercial butylated hydroxytoluene (BHT) since a lower concentration (100 ppm) provided a 90% DPPH scavenging effect.”
I am pretty sure you can add some discussion following this sentence regarding the negative side effects that synthetic antioxidants may generate and why even though the antioxidant effect of algae extracts is lower, it is much more beneficial to apply natural extracts to prevent those side effects.

L255: please remove "or lethal concentration (LC50)", it creates confusion.

Author Response

We have made the response to reviewer's comment in pdf file. Please see the attachment.

This manuscript is a resubmission of an earlier submission. The following is a list of the peer review reports and author responses from that submission.